# Interprofessional collaboration between hospital-based palliative care teams and hospital ward staff: A realist review

Louana Moons[1,2‡], Fouke Ombelet[3,4‡], Mieke Deschodt[1,5*], Maaike L. De Roo[1,6], Eva Oldenburger[2,7], Inge Bossuyt[2], Peter Pype[8]

**1** Department of Public Health and Primary Care, Gerontology and Geriatrics, KU Leuven, Leuven, Belgium, **2** Department of Palliative Medicine, University Hospital Leuven, Leuven, Belgium, **3** Department of Neurology, University Hospital Leuven, Leuven, Belgium, **4** Department of Neurosciences, Leuven Brain Institute (LBI), KU Leuven, Leuven, Belgium, **5** Competence Center of Nursing, University Hospital Leuven, Leuven, Belgium, **6** Department of Geriatrics, University Hospital Leuven, Leuven, Belgium, **7** Department of Radiation Oncology, University Hospital Leuven, Leuven, Belgium, **8** Unit Interprofessional Collaboration in Education, Research, and Practice, Department of Public Health and Primary Care, Faculty of Medicine and Health Sciences, Ghent University, Ghent, Belgium

‡ LM and FO are joint first authors.
* mieke.deschodt@kuleuven.be

## Abstract

### Introduction

To ensure high quality in-hospital palliative care delivery, we need a deeper understanding of the interprofessional collaboration processes between palliative care teams (PCTs) and hospital ward staff. This realist review aims to unravel the relationships between contextual (C) factors, mechanisms (M), and outcomes (O) related to this interprofessional collaboration process.

### Methods

This realist review employed an iterative, theory-driven approach which led to the development of a refined program theory explaining how, for whom, and under which conditions interprofessional collaboration between PCTs and hospital ward staff contributes to quality of care. On April 21st, 2023, five electronic databases were searched for relevant articles published between January 2013 and April 2023. Data screening and extraction was carried out by two independent researchers. Data-analysis was a two-phase iterative process to develop an overarching CMO-model and ground this model within third-generation cultural-historical activity theory (CHAT).

### Results

We identified 10 recurrent mechanisms contributing to the process of interprofessional collaboration and several moderating factors influencing ward staff's

**Data availability statement:** All relevant data are within the paper and its Supporting Information files.

**Funding:** This realist review manuscript was financed with internal funds provided by the Department of Palliative Medicine at University Hospital Leuven in Belgium. The funder did not have a role in study design, data collection and analyses, decision to publish, or preparation of the manuscript.

**Competing interests:** The authors have declared that no competing interests exist.

willingness to initiate PCT involvement. Contextual factors at both ward staff and PCT levels triggered mechanisms related to awareness, feelings, perspectives, and expectations, shaping outcomes for ward staff, PCTs, patients, and relatives. Notable outcomes included improved patient understanding of their disease, relatives' increased satisfaction with care, greater PCT involvement, and a sense of relief among ward staff. The CHAT-analysis revealed the interplay between expectations, roles, and beliefs in shaping interprofessional collaboration. Dual effects of mediating artifacts, such as palliative care training and time constraints, were observed depending on context, further illustrating the complexity of collaborative practices. Ultimately, this realist review emphasized the need for hospital decision-makers to acknowledge the multidimensional aspects of palliative care and foster a sense of partnership between PCTs and ward staff to optimize IPC. A refined program theory was developed to guide future interventions.

## Conclusion

This realist review highlights the complexity of interprofessional collaboration between PCTs and ward staff, emphasizing the importance of tailored approaches that address specific contextual needs, expectations, and norms. Strengthening positive attitudes, clarifying roles, and fostering partnerships can enhance interprofessional collaboration, ultimately improving palliative care quality in hospital settings.

## Background

Due to demographic shifts, such as an ageing population and a growing prevalence of non-communicable diseases, the demand for palliative care (PC) is expected to grow substantially [1,2]. It is estimated that by 2060 nearly half of all deaths globally will involve patients experiencing serious health-related suffering requiring PC [3]. Patients with non-communicable diseases also face higher hospital readmission rates and increased PC needs as their condition progresses [4–6]. Therefore, we need to ensure high-quality in-hospital PC delivery worldwide.

In many Western countries, specialized palliative care teams (PCTs) play a crucial role in hospitals to integrate PC principles into acute care wards [7,8]. PCTs are multidisciplinary, often including physicians, nurses and psychologist. Rather than taking over primary patient care, PCTs provide consultative support and advise to hospital ward staff regarding palliative care. For example, in a case of a patient with advanced cancer and escalating symptoms, the PCT might support hospital ward staff in advance care planning conversations, adjusting medication prescription, and coordinating family support, while hospital ward staff remains responsible for daily care delivery. This underscores the importance of effective interprofessional collaboration (IPC) to ensure optimal PC delivery [9].

We define IPC as the collaborative practice among professionals from diverse backgrounds in which shared accountability, interdependence, and clarity of roles

are of importance to deliver the highest quality of care to patients, their families, and carers [10–12]. Evidence suggests IPC between PCTs and hospital ward staff can be enhanced by recognizing mutual expertise, facilitating communication, clarifying roles, and providing education opportunities and continuous support [7,9]. However, significant variability exists in collaborative models, and their impact on patient outcomes remains unclear, partly due to a limited understanding of PCTs' operational practices. [9,13].

To optimize the quality of in-hospital PC, we need a deeper understanding of these interprofessional collaborative processes and their mediating factors. Therefore, this realist review aimed to provide insights into how, for whom and under which conditions IPC between PCTs and ward staff of acute care hospital wards contributes to the quality of care for everyone involved (e.g., patient's quality of life, care team satisfaction, cost efficiency) [14,15].

## Methods

A realist review is a theory-driven approach to evidence synthesis that seeks to understand how, why, and under which conditions specific interventions or processes work [16]. The complete methodology is reported in our protocol paper [12]. We followed the methodological guidance by Pawson and colleagues and findings are reported in accordance with the Realist And Meta-narrative Evidence Synthesis: Evolving Standards (RAMESES) quality and publication standards [17,18]. To initiate the review, we developed a preliminary program theory offering first insights into how IPC was suspected to work according to existing literature (see supplementary file 1 (S1 File)). The research team included academic experts in implementation science (MD), nursing research (LM), palliative care research (MDR), and realist review methodology and interprofessional collaboration (PP) as well as clinical experts in palliative care (EO, IB).

### Search for evidence

A search was conducted across five electronic databases: MEDLINE (as searched by PubMed), Embase, CINAHL, Web of Science, and Scopus on April 21st, 2023. Full details of all search syntaxes can be found in supplementary file 2 (S2 File). We limited our search to articles published within the last ten years (2013–2023). Grey literature was not considered, as a considerable number of papers covering a sufficient breadth of study designs and ensuring a comprehensive coverage of our topic was identified.

### Screening and selection

Two independent researchers (LM, FO) screened articles by title and abstract, followed by full-text screening based on in- and exclusion criteria. Ultimately, we included studies with a qualitative, quantitative (experimental and non-experimental), or mixed-method design, published in English or Dutch, and conducted in high-income countries based on the World Bank country classifications in 2023 [19]. We excluded studies conducted outside of acute care hospital wards, studies involving pediatric populations (<18 years) and studies primarily focusing on the COVID-19 pandemic. For interventional studies, a clear description of the collaborative interactions between PCTs and hospital ward staff was required. Non-interventional studies were included if they provided relevant insights contributing to our research question and the development of our refined program theory. Studies highlighting facilitators and barriers to PCT involvement were included to explore moderating factors related to IPC.

### Extraction and appraisal

Relevant study data were extracted by two independent researchers (LM, FO) into standardized data extraction forms (supplementary file 3 (S3 File)) and appraised for their fitness for purpose based on their *relevance*—assessed by the extent to which they contribute to the theory refining process—and their *rigour*, which evaluates the credibility and trustworthiness of the methods used to generate the relevant data. Relevance was scored as high, medium or low, while rigour was scored as either high or low.

## Analysis and synthesis

**Phase 1: Constructing CMO-configurations.** Key contexts (C), mechanisms (M), and outcomes (O) were identified in each article to construct individual CMO-configurations. 'Contexts' are the conditions necessary for a mechanism to be triggered. 'Mechanisms' are the processes occurring within individuals and driving change in order to generate outcomes of interest. Lastly, 'outcomes' encompass all relevant effects of IPC between PCTs and hospital ward staff, observed at the levels of patients, relatives, care workers, and hospitals [20].

High-rigour articles (n = 22) were analyzed first and individual CMO-configurations were extracted. Thereafter, individual CMO-configurations were clustered by grouping similar mechanisms. Ultimately, recurrent mechanisms were combined which led to an overarching CMO-model. Regarding the low-rigour articles (n = 26), only those with high (n = 1) or medium (n = 9) relevance were analyzed to strengthen the CMO-model. Articles with both low relevance and low rigor (n = 16) were excluded from the analysis as they were deemed not to provide any additional rigorous information to refine the CMO-model.

**Phase 2: Grounding CMOs within third generation cultural historical activity theory (CHAT).** Identified recurrent mechanisms from phase 1, with their corresponding CMO-configurations, were analyzed using third-generation CHAT as an analytical framework. This way, the culture wherein the interprofessional collaborative process occurs is examined in order to identify appropriate strategies to optimize this process.

Third generation CHAT is a social theory suggesting that multiple activity systems can interact with one another [21–23]. Each activity system consists of 6 components interrelated with each other and the activity being carried out:

1. the goal of the activity (*object*).

2. an individual with its own perspectives and emotions (*subject*).

3. resources, protocols, or tools (*mediating artifacts*).

4. policies, norms, and expectations (*rules*).

5. social and relational dynamics (*community*).

6. division of tasks and roles (*division of labor*).

Using CHAT's third generation enables us to focus on the interactions taking place between a PCT *(activity system 1)* and a hospital ward *(activity system 2),* how they may develop shared objectives, encounter moments of friction, and develop learning opportunities.

The analytical phase involved extensive round table discussions with experts in PC, ward staff, and researchers, in which every CMO-configuration was evaluated and grounded within CHAT third-generation by connecting the CMOs from phase 1 to the CHAT components (object, subject, mediating artifacts, rules, community, and division of labor). Additional discussion rounds were conducted with the entire research group to refine interpretations and ensure alignment. This enabled us to unpack contradictions as well as synergies within and between both activity systems. Overall, phase 2 contributes to the theory-building process within this review where the end-goal was to deliver a refined program theory providing insights into how interprofessional collaboration currently works, which important contradictions exist and where learning opportunities can be found.

## Results

A total of 5,658 unique articles were identified and screened by title and abstract, leading to a full-text analysis of 266 articles based on the in- and exclusion criteria. Ultimately, 48 papers were included in this review, of which 32 were used in the analytical phase. Fig 1 illustrates the entire screening process.

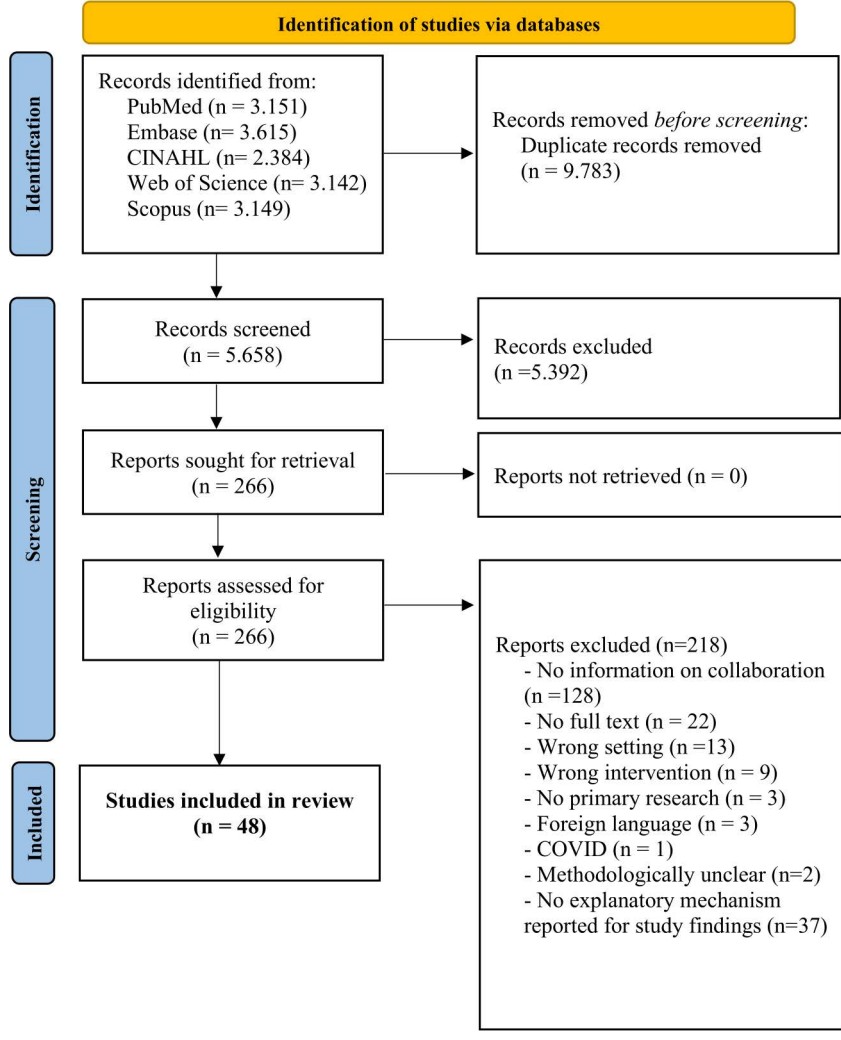

**Fig 1. Screening process.**

## General characteristics of included studies

Supplementary file 4 (S4 File) provides an overview of the general characteristics of the 48 included papers. Of these, 24 (50.0%) came from the USA, 15 (31.3%) from Europe, 8 (16.7%) from Asia, and one study was conducted in Canada.

We included 28 quantitative, 15 qualitative and five mixed-methods studies. Twelve (25%) studies were conducted within an oncology department, 15 (31.3%) studies included different hospital departments, 5 (10.4%) studies were conducted within an intensive care unit. Other studies included the following hospital wards: hematology (n = 2), geriatrics (n = 1), hepatology (n = 1), internal medicine and surgical care (n = 1), oncology and palliative medicine (n = 1), and pneumology (n = 1). Nine (18.8%) studies did not specifically report the type of hospital ward.

All included studies comprised a variety of participant groups. Most studies focused exclusively on one group, such as patients (n = 17) or ward staff (n = 17). Others included combinations of groups, such as ward staff and PC specialists (n = 8), patients and ward staff (n = 3), or all three key stakeholder groups (n = 2). Only one study also included relatives. Most study interventions focused on PC consultations, services or programs consisting of a PCT with different

PC specialists (e.g., physicians, nurses, social workers). Solely two studies focused on the involvement of only one PC specialist: a clinical nurse specialist and a PC specialist social worker.

## Phase 1: Constructing CMO-configurations

During this phase, we identified several moderating factors that appeared to be prerequisites needed for collaboration to occur. Therefore, we did not include these in our overarching CMO-model but describe them separately as factors both disabling and enabling ward staff's motivation to initiate PCT involvement. We identified moderating factors at ward staff, PCT, and patients and relatives' level, as summarized in Table 1.

At ward staff level, moderating factors related to the staff's knowledge and skills in PC *(e.g., a lack of understanding about the role of PC)*, their attitudes, beliefs, and expectations regarding PC *(e.g., not believing PC can offer a benefit)*, and their perceptions of the PCT, patients, and relatives *(e.g., perception PCT seems busy)* [24–31]. Moderating factors at the PCT level included responding in a timely manner, having unclear roles and responsibilities, having a lack of human resources, and certain behaviors such as not involving ward staff in decision-making or being too aggressive in recommending treatment withdrawal [26,27,32,33]. Lastly, moderating factors related to patients and relatives included resistance to PC, relatives being less involved in patient's care, or patients hoping for cure [26,27,33].

Following this, the analysis of 32 included articles identified 54 distinct CMO configurations, as detailed in supplementary file 5 (S5 File). From these, 10 recurring mechanisms emerged, which are outlined in supplementary file 6 (S6 File). These mechanisms were synthesized into an overarching CMO model, presented in Fig 2 and explained below.

**Ward staff's awareness for PC or PC services.** Ward nurses reported that (biweekly) case discussions enhanced their awareness of all dimensions of PC, thereby facilitating multidimensional symptom management for patients [34]. Additionally, PC training for ward staff positively influenced awareness of PC organization in the hospital [35]. Conversely, high staff turnover diminished ward staff's awareness for the PCT [24]. The importance of the PCT's presence and visibility was highlighted in two studies [24,33]. For instance, regular visits by the PCT to hospital wards or actions that

**Table 1. Moderating factors influencing ward staff's motivation to initiate PCT involvement.**

| | Disabling moderating factors | Enabling moderating factors |
|---|---|---|
| **Ward staff** | • Lack of time [24]<br>• Feeling resentment towards PC among relatives [26]<br>• Perception that PCT seems busy [26]<br>• Expecting treatment and care by PCT not to be in their preferred way [27]<br>• Not receiving support as desired [27]<br>• Taking pride in own symptom management skills [27]<br>• Taking ownership over the patient and the treatment [27]<br>• Only considering treatment as an option and focusing on intent-to-cure treatment [27]<br>• Being concerned about patient's negative image of the PCT [27]<br>• Hoping for a cure [27]<br>• Having perceptions that patients do not understand what PC is [28]<br>• Not believing PC can offer a benefit [29]<br>• Thinking it is not their place to suggest a PC consultation [29]<br>• Having a lack of understanding about the role of PC [32]<br>• Having confidence in own abilities to deliver good PC [32] | • Lack of time [26,30]<br>• Trust [25]<br>• Having a lack of PC skills [26]<br>• Noticing a knowledge gap in patients about their disease trajectory [27]<br>• Sensing intra-family disagreements [27] |
| **PCT** | • Unclear roles and responsibilities [33]<br>• Insufficient human resources within PCT [27]<br>• Not involving ward staff in decision-making [31]<br>• Being too aggressive in treatment withdrawal [31] | • Responding in a timely manner [26] |
| **Patients and relatives** | • Patients and relatives confused about or showing resistance to PC [33]<br>• Relatives are less involved in patient care [26]<br>• Hoping for a cure [27] | |

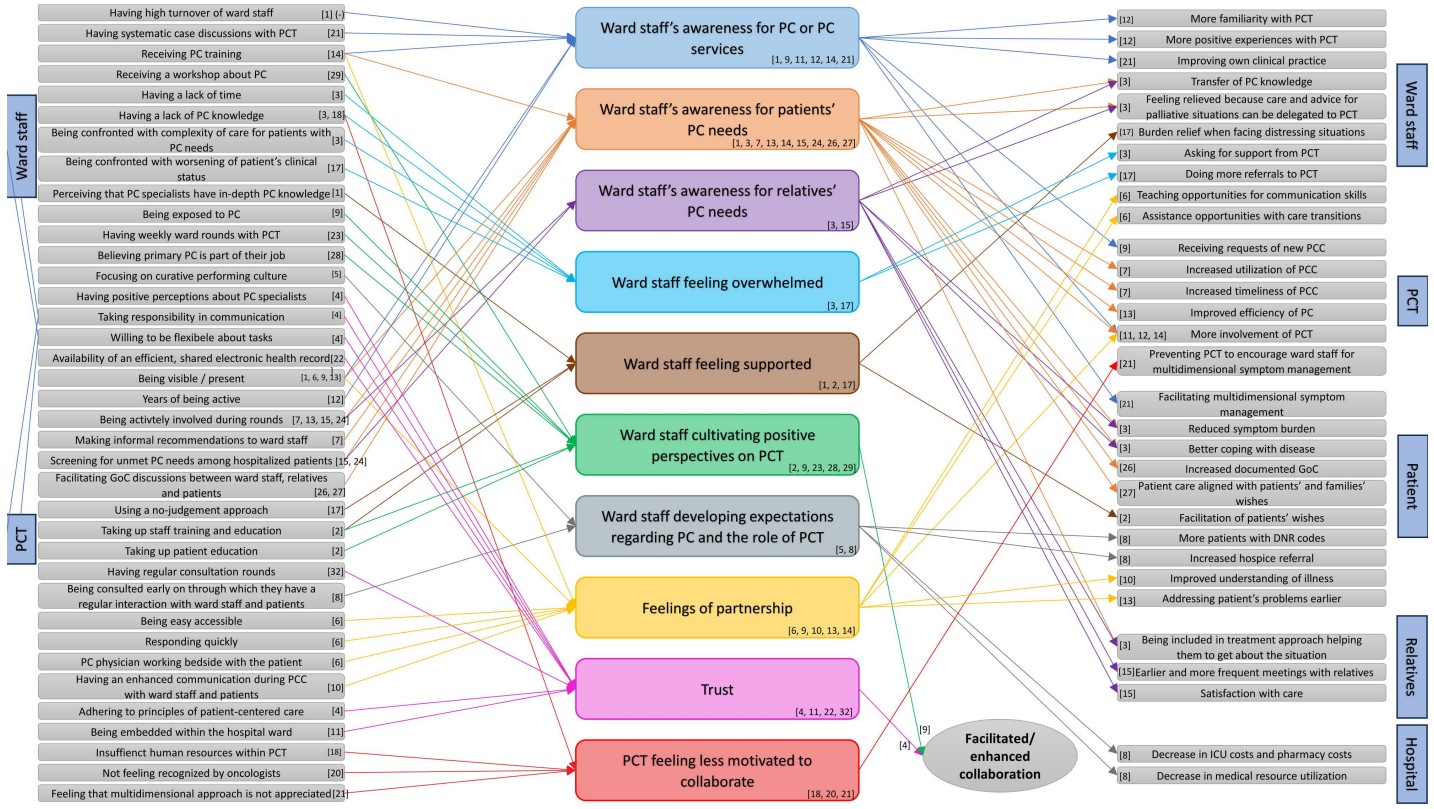

**Fig 2. Overarching CMO-model.** PC: Palliative Care, PCT: Palliative Care Team, PCC: Palliative Care Consultation, GoC: Goals of Care, DNR: Do-Not-Resuscitate, ICU: Intensive Care Unit.

enhanced its visibility were found to trigger ward staff's awareness of PC and its services [24,33]. Over time, ward staff becomes aware of the added value of a PCT, fostering greater familiarity and positive experiences with its services [36]. Reported outcomes included improved clinical practice among ward staff [34], increased involvement of the PCT in patient care [35–37], and the requests of new consultations [33] once awareness for PC or the PCT was present.

**Ward staff's awareness for patients' PC needs.** Four articles described PC interventions where PC specialists actively participated in morning rounds on hospital wards. This involvement triggered ward staff's awareness of patients' PC needs [38–41]. One study noted that this active involvement created opportunities for PC specialists to make informal recommendations to ward staff regarding PC, further triggering an increased awareness of patients PC needs [38]. Additional contextual factors were identified as triggering the awareness for patients' PC needs, including a PCT being present through regular ward visits, screening for unmet PC needs in hospitalized patients, and PCTs facilitating goals-of-care discussions [24,40–43]. Eventually, receiving PC training was found to cause awareness for patients' PC needs among ward staff [35]. One study reported that ward physicians, once aware for patient's PC needs, became motivated to involve PC specialists. This collaboration allowed the specialists to provide advice and support, resulting in knowledge transfer, and gave ward physicians a sense of relief as they could delegate aspects of patient care to the PCT [44]. Awareness for patients' PC needs led to several positive outcomes: reduced symptom burden for patients, improved coping with their disease, and relatives being included in the treatment approach enhancing their understanding of the situation [3]. At the PCT level, reported outcomes included increased utilization and timeliness of PC consultations, improved efficiency of PC, and increased involvement of the PCT in patient care [35,38,39].

**Ward staff's awareness for relatives' PC needs.** Two included articles reported on a mechanism related to relatives. The processes that enhance ward physicians' awareness of patients' PC needs also raise their awareness for relatives' PC needs [44]. Furthermore, active involvement of a PCT during rounds or PC specialists screening for unmet PC needs in patients triggers ward staff's awareness for relatives' PC needs [40]. This contributes to earlier and more frequent family meetings and improved relatives' satisfaction with care [40].

**Ward staff feeling overwhelmed.** Two studies identified several contextual factors contributing to ward staff feeling overwhelmed, including lack of time, insufficient PC knowledge, the complexity of caring for patients with PC needs, and being confronted with worsening of the patient's clinical status. These challenges often prompted ward staff to seek support from the PCT or make more referrals to them. Consequently, when ward staff feels overwhelmed, they are more inclined to involve the PCT [44,45].

**Ward staff feeling supported.** Positive perceptions of PC specialists among ward staff played a significant role in fostering feelings of support. For instance, recognizing the extensive knowledge of PC specialists triggered ward staff to feel supported [24]. Similarly, education provided by a PC clinical nurse specialist, equipped ward staff with the tools and guidance to facilitate patients' wishes [46]. Additionally, PC specialists maintaining a non-judgemental approach made ward staff feel supported, leading to burden relief when facing distressing situations [45].

**Ward staff cultivating positive perspectives on PCT.** One study showed that weekly ward rounds with the PCT positively influenced ward staff's perceptions of the PCT [47]. Additionally, physicians who attended a workshop about PC recognized their PCT as useful [48]. Furthermore, when physicians believe PC is part of their job they are able to identify the benefits of a PCT [30]. Moreover, PC specialists engaging in patient education or providing staff education and training triggers ward staff to cultivate positive attitudes and a good acceptance of the PC specialists' role [46]. In one study, PC specialists reported exposure to PC as an important factor for cultivating positive views about the PCT enhancing collaboration [33]. However, this statement was not explicitly confirmed by the perspective of ward staff in this publication.

**Ward staff developing expectations regarding PC and the role of PCT.** Ward staff that prioritizes active, discipline-specific treatment, experience a lack of time for PC, which results in the expectation that the PCT should provide direct guidance on decision-making related to PC [49]. Early PCT consultation (within 48 hours of admission) fostered regular interaction among the PCT, ward staff, and patients, which helped establish appropriate expectations and facilitated goals-of-care discussions. Eventually, this resulted in more Do-Not-Resuscitate (DNR) orders, increased hospice referrals, reduced Intensive Care Unit (ICU) and pharmacy costs, and lower medical resource utilization [50].

**Cultivating feelings of partnership.** Awareness of PC organization within the hospital and a better comprehension of specific roles, was enhanced in ward staff that received a 4-hour PC training session. This process cultivated a sense of partnership which facilitated PCT involvement [35]. Accessibility of the PCT, their quick responsiveness, and their bedside engagement with patients, also triggered feelings of partnership between the PCT and ward staff [51]. This partnership led PC specialists to teach ward staff communication skills and assist with care transitions [51]. Furthermore, visible collaboration between the PCT and ward staff promoted early relationship-building between PC specialists and patients and their families, enabling them to address patient's problems sooner [39]. Two studies found that PCT visibility in general or their presence during morning rounds cultivated feelings of partnership [33,51]. Enhanced communication between patients, PC specialists, and ward staff during PC consultations also contributed to cultivating feelings of an effective partnership which improved oncologists' skills and comfort in communication, ultimately enhancing patients' understanding of their illness [52].

**Fostering trust.** Embedding a PCT within a hospital unit or conducting regular consultation rounds fosters feelings of trust among ward staff [37,53]. Trust was also fostered when ward social workers had positive perceptions of PC social workers' abilities and when PC social workers adhered to principles of patient-centered care [25]. This study identified two other mechanisms interrelated with trust, namely information-sharing and role-negotiation. Contextual factors directly triggering these mechanisms were taking responsibility in communication and willingness to be flexible about tasks

[25]. Trust, along with effective information-sharing and role-negotiation, was critical for successful interprofessional collaboration between ward and PC social workers, while the absence of any of these elements hindered it [25]. Additionally, an efficient, shared electronic health record system facilitated timely, relevant, and complete information exchange [54].

**PCT feeling less motivated to collaborate.** Challenges to collaboration were reported when PC specialists perceived ward staff to have insufficient PC knowledge [27]. Furthermore, in this study, due to staffing problems within the PCT, hematologists reported that PC specialists used a less active approach towards them [27]. Additionally, when PC specialists felt unrecognized or believed their multidimensional approach was not appreciated, they found collaboration more challenging [28,34]. In the latter, this prevented the PCT to encourage ward staff in multidimensional symptom management [34].

**Remaining individual CMO-configurations.** Not all CMO-configurations derived from individual articles could be integrated in the overarching CMO-model. Some contextual factors are similar, but they trigger different mechanism and outcomes and are therefore described separately.

A cross-sectional study with registered nurses found that those working at oncology or hematological wards exhibited more favorable attitudes toward their PCT and demonstrated a greater understanding of the consultation process. This enhanced understanding was also observed among nurses who had frequent interactions with the PCT or had received prior education in PC. In addition, they showed improved abilities regarding collaboration with their PCT *(e.g., better pro-active communication with PCT)* [55].

The presence of a PCT, whether through consultation or integration within the care team, facilitated shared decision-making, which in turn contributed to holistic patient care, improved pain management, an increase in DNR orders, as well as more hospice transfers [37,43]. Additionally, having a PC specialist integrated within the hospital wards' care team provided access of expertise, through which management of broader patient problems is facilitated [39]. Furthermore, PCT presence by regularly visiting the hospital wards influenced ward staff's need for more PC knowledge [24]. Moreover, when ward staff is aware of deficits in their PC knowledge, they are more receptive to learning during interaction with the PCT [51].

Lastly, using a co-rounding model fostered more direct communication, contributing to greater clarity, mutual understanding, and respect between teams. Ultimately, leading to greater efficiency in delivering PC [39]. However, disagreements between teams were perceived as inhibiting efficient care delivery [39]. Additional challenges, such as busy schedules, failure to act on PCT's suggestions as well as differing views on PC and how good PC is achieved, contributed to misunderstandings, frustrations and suboptimal communication [27,49].

## Phase 2: Grounding CMOs within third-generation CHAT

Fig 3 depicts the refined program theory of this realist review. It captures the interrelations among all CHAT components within and between the activity systems of a PCT and a hospital ward, providing insights into how interprofessional collaboration currently works between both. A detailed summary of the specific results from phase 1 within CHAT can be found in supplementary file 7 (S7 File).

The following section presents our findings grounded within the CHAT components: object, subject, mediating artifacts, rules, community, and division of labor.

**Object.** In the **PCT activity system**, the primary object is to ensure that every in-hospital patient with PC needs receives appropriate PC *(Object 1)*. If interaction with the hospital ward activity system occurs, the objective of the PCT is to enable ward staff to provide PC independently *(Object 2)*. PCT involvement contributes to *Object 2* by providing the necessary support, advice, and expertise to ward staff, thereby strengthening their ability to deliver optimal PC. Additionally, the sense of partnership between ward staff and PCT members fosters opportunities for interpersonal learning, enabling PCTs to teach and assist ward staff in developing their PC skills. This too contributes to ward staff being

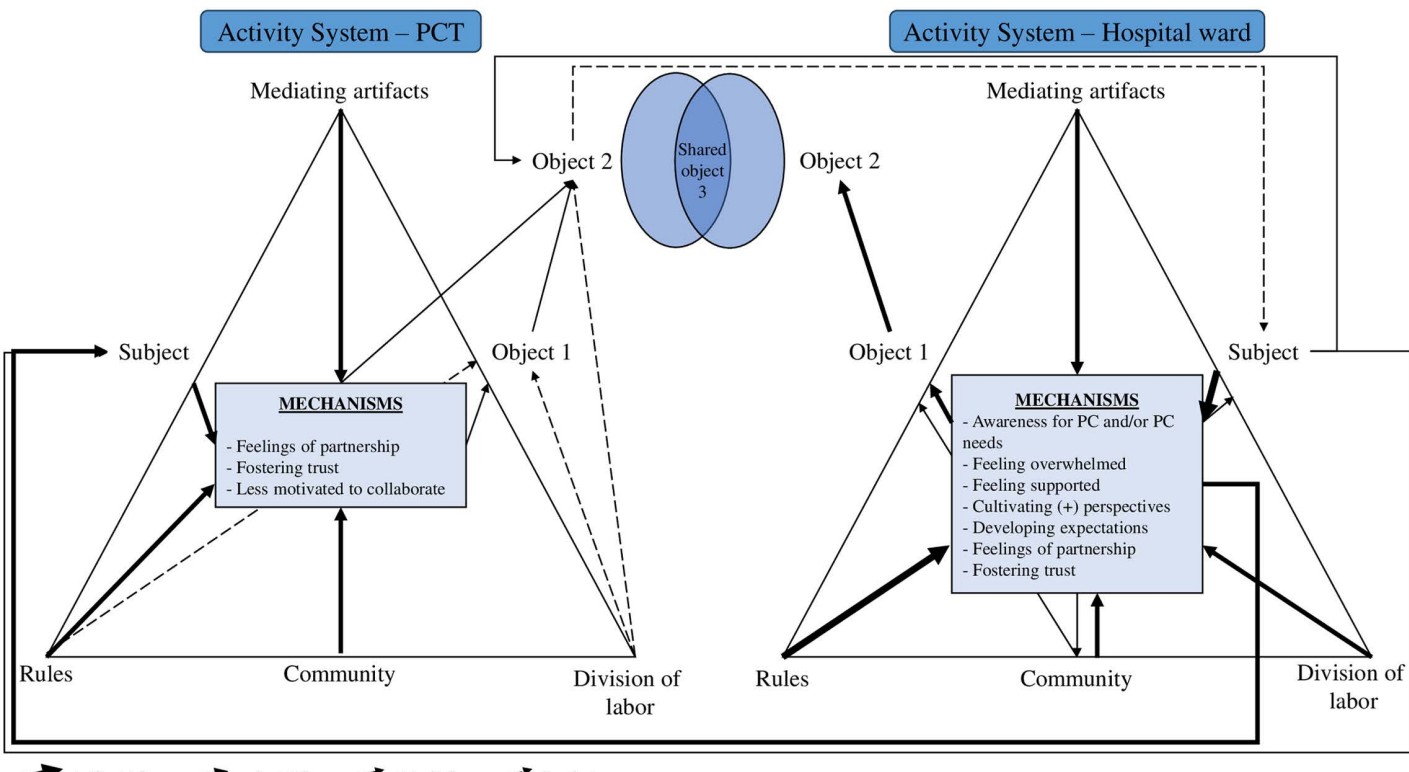

**Fig 3. Refined program theory.** PCT: Palliative Care Team, PC: Palliative Care.

able to deliver PC independently. However, a lack of motivation among PC specialists to collaborate creates friction, as it inhibits the application of a multidimensional approach to patient care. This, in turn, prevents the PCT from achieving its second object of enabling ward staff to provide PC, impeding both effective interprofessional collaboration and the provision of optimal care.

Within the **hospital ward activity system**, *Object 1* focuses on delivering discipline-specific high-quality care for every patient. If interaction with the PCT activity system occurs, *Object 2* is to ensure that all patients on the ward receive optimal PC. When ward staff feels supported by the PCT, this reduces ward staff's burden and distress, leading to outcomes in line with the ward's objectives, such as reduced patients' symptom burden, improved patient coping, and better inclusion of relatives in treatment plan. Furthermore, when appropriate expectations are set through goals-of-care discussions, ward staff is more likely to refer patients to the PCT, resulting in improved patient outcomes, such as more hospice referrals, and reduced ICU and pharmacy costs. These actions directly contribute to the hospital ward's objectives one and two of providing holistic care and optimal PC.

**Subject.** Within the hospital ward activity system, ward staff are central subjects whose internal states, such as their perceptions, beliefs, knowledge, and professional attitude significantly influence their engagement with the PCT. For example, when ward staff perceive the PC specialists as highly knowledgeable, they feel supported, which may enhance their motivation to collaborate. In addition, a lack of PC knowledge among ward staff triggers feelings of being overwhelmed, creating a need for support from the PCT. Ward staff's beliefs about the scope of their role, for instance considering PC as their responsibility, enables them to develop positive perspectives towards the PCT. In contrast, ward staff prioritizing a treatment-focused culture can lead to the expectation for the PCT to direct their actions. This dynamic

inhibits the PCT's objective of enabling ward staff to provide PC independently, creating friction between the two activity systems.

From the perspective of PC specialists, one study highlighted that when PC social workers adopted a patient-centered care approach, ward social workers perceived them more positively. This positive perception fostered trust, making collaborative relationships more likely to develop. On the other hand, if PC specialists feel their way of working is not recognized or valued by ward staff, it can demotivate them and reduce their willingness to collaborate. Thus, the internal states and motivations of subjects in both activity systems – shaped by their perceptions, knowledge, and beliefs – are central in determining the quality of interprofessional collaboration and in achieving the second objectives of both the hospital ward and the PCT.

**Mediating artifacts.** Mediating artifacts play a crucial role within and between the activity systems of a hospital ward and a PCT, influencing how subjects behave, think, and collaborate. One such artifact is **PC training**, serving as both a catalyst and inhibitor for collaboration. Several studies report on the profound impact of PC training on ward staff's awareness of PC services and patients' PC needs. PC training can foster positive perceptions of the PCT, creates a sense of synergy between both activity systems, and enhances ward staff's motivation to involve the PCT in patient care. The latter of which creates a positive feedback loop, as greater involvement of the PCT generates more interaction and opportunities for learning, further reinforcing collaboration. However, the impact of PC training is not uniform, with several studies reporting conflicting results related to PC training and the level of PC skills among ward staff. For instance, both ward staff feeling confident in their own PC skills *(e.g., taking pride in their symptom management skills)* and those that lack PC knowledge *(e.g., limited understanding of the role of PC)* may be less inclined to involve their PCT. Conversely, in other studies, ward staff with limited PC skills may actually be more motivated to seek PCT involvement. Another mediating artifact, **lack of time**, triggers similarly mixed responses: some staff, feeling overwhelmed by the lack of time do seek PCT involvement, while others are less motivated to initiate collaboration. These dual effects, highlight the complexity of mediating artifacts in shaping interactions between PCTs and hospital wards.

Mediating artifacts such as **communication tools** like shared electronic health records facilitate information sharing between ward staff and PC specialists, possibly fostering trust and a sense of partnership. Meanwhile, a **lack of human resources** within the PCT limits the team's ability to actively approach and engage with ward staff, further complicating collaboration efforts.

**Rules.** The presence of certain **norms**, such as PCT involvement through regular consultation rounds or presence of the PCT during ward rounds, play a critical role in fostering trust and a sense of partnership between PCTs and ward staff. Regular PCT involvement enhances ward staff's awareness of PC needs and services through which knowledge transfer between the systems is facilitated. This creates a positive feedback loop, because ward staff's awareness reinforces engagement with the PCT. This aligns with the PCT's goal of enabling ward staff to provide PC independently and the hospital ward's aim of delivering optimal PC. Furthermore, when ward staff hold negative **expectations** about the care and treatment provided by the PCT, their motivation to engage decreases. Similarly, when ward staff expect to be involved in decision-making but perceive exclusion by the PCT, their willingness to consult the PCT diminishes. These misalignments in expectations create friction between the systems, undermining effective collaboration.

**Community.** In the hospital ward activity system, the community includes PCTs, patients, and their relatives, all of whom are vital to the mechanisms that drive interprofessional collaboration. For example, ward staff engaging in goals-of-care discussions with **patients** increases their awareness of patients' PC needs, often resulting in greater PCT involvement. In some instances, being confronted with the worsening of a patient's clinical status or the complexity of a patient's care caused ward staff to feel overwhelmed, triggering more referrals to the PCT. This process does not necessarily reflect collaborative interaction between the two systems, as it is primarily reactive rather than collaborative. Interactions with the community were often influenced by other components of the hospital ward's activity system, such as the beforementioned rules on PCT involvement. Conversely, other factors within the community demotivate ward staff

from engaging the PCT, such as limited **involvement of relatives** in patient care, ward staff feeling resentment towards PC among patients and relatives, or patients hoping for a cure. On the other hand, sensing a lack of understanding about the disease trajectory among patients or sensing intra-family disagreements are moderating factors positively influencing the initiation of PCT involvement.

Within the PCT activity system, the community factor of ward staff's PC knowledge plays a significant role. A lack of PC knowledge among ward staff can reduce PC specialists' motivation to collaborate. Furthermore, when PCTs are perceived as having in-depth PC expertise, ward staff feel supported, which strengthens the partnership between the two systems.

**Division of labor.** Our review demonstrates the importance of having **well-defined tasks** for interprofessional collaboration to occur. For example, when a PCT has the task to provide PC education to ward staff, it shifts the dynamics within the hospital ward, making the ward staff feel more supported. This educational task also triggers ward staff to cultivate positive perspectives on the PCT, recognizing the team as beneficial and accepting the specialist's role. In addition, the division of labor underscores the need for flexibility in task assignments to enable role negotiation.

Furthermore, our CHAT analysis demonstrates the need for **clear role definitions**. For example, screening for unmet PC needs was a PCT-role which directly influenced ward staff's awareness of patients' and families' PC needs. However, when roles are unclear this reduces ward staff's motivation to involve the PCT. The lack of role clarity may lead to misunderstandings about responsibilities, resulting in ward staff feeling unsure of when and how to consult the PCT. The perception that it is not ward staff's place to suggest a PC consultation further reduces engagement and collaboration. Therefore, a clear delineation of tasks and roles is essential to ensure trust and meaningful partnership between ward staff and PCTs.

## Discussion

This realist review provides insights into how, for whom and under which conditions IPC between PCTs and ward staff of acute care hospital wards contributes to the quality of care for everyone involved based on 52 individual CMO-configurations revealing 10 recurrent mechanisms. Various contextual factors at both the ward staff and PCT level were found to trigger mechanisms related to ward staff's awareness, feelings, perspectives, and expectations, as well as PCT's motivation to collaborate. These mechanisms influence several outcomes at the levels of ward staff, PCTs, patients, and relatives. Notable outcomes included patients' improved disease understanding, relatives' increased satisfaction with care, greater PCT involvement, and a sense of relief among ward staff. Using the CHAT third-generation theory, the importance of expectations and norms *(rules)* regarding PCT involvement and individual perceptions and beliefs *(subject)* about one another were highlighted. All these factors both positively and negatively influence the IPC process. The complexity of the collaborative practices between PCTs and ward staff, was further highlighted by the dual effects of PC training and time constraints depending on its context. Ultimately, a refined program theory was developed.

### Frictions, contradictions, and learning opportunities

To identify organizational learning opportunities within the context of interprofessional collaboration in PC, it is essential to interpret moments of friction within or between the different activity systems. To do so, we draw on the concepts of single-loop and double-loop learning. Single-loop learning involves addressing problems by adjusting actions within existing frameworks, norms, or strategies, while double-loop learning goes deeper, challenging and revising underlying organizational values and norms to drive transformative change [56,57].

For instance, following the interprofessional interaction, a PCT typically aims to enable ward staff to provide PC themselves. This is in contrast with the goal of the ward staff, who typically engages with the PCT to ensure PC is delivered to its patients, regardless of who provides it. This highlights distinct but interdependent goals between the systems once they interact, which may hinder the alignment needed to achieve a shared objective. This moment of friction, however, may conceal valuable learning opportunities once both systems become aware, understand, and address each other's needs. For example, a PCT learning about ward staff's needs concerning patient care and ward staff gaining insights into long-term benefits

of PC training could lead to a mutual understanding about each other's goals through which single-loop learning is accomplished which might be essential for fostering effective IPC and advancing integrated PC delivery across hospital wards.

Friction also arises when staff is confronted with a patient's worsening clinical status but are constrained by lack of time. However, this perceived time pressure may be rooted in limited PC knowledge, leaving ward staff to feel overwhelmed, causing them to ask support from their PCT—a response characteristic of single-loop learning. To foster double-loop learning, hospitals should address these underlying causes of experiencing lack of time due to limited PC knowledge by investing in systemic solutions, for example a yearly PC training for ward staff. Such initiatives can gradually shift norms within the system and increase comfort with PC tasks among ward staff, reducing the sense of being overwhelmed even if the workload remains unchanged. While this approach may initially require additional effort, it can enhance efficiency and interprofessional collaboration (IPC) in the long term.

Furthermore, when priority is given to a treatment-focused culture ward staff experience a lack of time for PC and expect the PCT to direct their actions. This dynamic inhibits the PCT's objective of enabling ward staff to provide PC independently, creating friction between the two activity systems. To address this moment of friction, double-loop learning would involve a more profound reflection on the curative culture and its impact on optimal PC delivery. A mixed-methods study on generalist palliative nursing care revealed limited integration of PC in hospital wards due to organizational issues, such as the lack of a hospital-wide policy for general PC delivery [58]. Similarly, we urge hospital decision-makers to address the interaction between organizational culture and PC integration to enhance IPC and improve access to generalist PC for hospitalized patients. This could lead to changes in organizational policies supporting a more integrated approach to patient care.

A key mechanism facilitating IPC is ward staff's awareness of PC, the available PC services, and the PC needs of patients and their families. This awareness motivates ward staff to involve the PCT in patient care, leading to improved patient outcomes (e.g., reduced symptom burden, better coping) and knowledge transfer, in turn enhancing ward staff's ability to provide PC independently. To foster effective IPC, hospitals should prioritize actions that build awareness of PC among ward staff. One such action, highlighted in our review, is PC training. A quasi-experimental study further demonstrated that PC training significantly improves nurses' perceived self-efficacy, psychosocial support skills, and symptom management capabilities [59]. Hospitals should aim at redirecting their organizational norms towards more PC training for ward staff in order to facilitate the IPC process and optimize PC delivery across hospital wards.

Lastly, cultivating feelings of partnerships is another key mechanism for IPC to occur. When partnership is established, ward staff are more likely to involve PC specialists in patient care, leading to teaching opportunities, assistance, and earlier identification of patient problems. An important factor in fostering partnership is the presence of PC specialists during morning rounds. Hospitals not yet prioritizing this approach should consider restructuring their PCT's functioning to promote effective IPC and ensure high-quality PC delivery.

## Methodological considerations

This review was conducted by an interdisciplinary team incorporating diverse professional perspectives and providing expertise in different research and clinical areas. The interventions described in the studies were often multi-component and heterogeneous, making it challenging to identify the specific components that were responsible for the observed changes. Therefore, many interventional studies had to be excluded due to the lack of identified explanatory mechanisms. This highlights the need for researchers to focus more on explaining how interventions work, rather than just listing their components [60,61], for example, by the use of intervention reporting guidelines such as the Template for Intervention Description and Replication (TIDieR) Checklist [62]. Hence, providing detailed insights into the mechanisms behind interventions can help inform future research and practice, offering guidance on strategies that could be adapted to different contexts. Furthermore, the literature search underlying this realist review was completed over a year ago, reflecting the extensive and iterative nature of theory-building in realist methodology The large volume of included studies

and the multiple rounds of analysis and discussion among the research team were essential to constructing a robust and context-sensitive program theory. Given the specificity and maturity of this theory, we consider it unlikely that recently published studies would substantially change our conclusions.

### Gaps in the evidence

We identified several important gaps in the literature. First, we observed limited evidence regarding the PCT's perspective, resulting in fewer interactions within the PCT activity system. Additionally, the relatives' role within the IPC progress was not well described: only two studies focused on relatives, highlighting the importance of including their perspectives in future research. Insufficient evidence was observed regarding the division of roles and tasks with few studies detailing the specific tasks and roles of PCTs and ward staff. Also limited evidence was found regarding the description or evaluation of existing protocols *(rules)* about the functioning of PCTs or their collaboration process with ward staff. Lastly, it seems that limited research has been conducted beyond the perspective of physicians and nurses resulting in minimal evidence on other ward staff's perspectives such as psychologists, social workers, or chaplains. Furthermore, greater attention should be given to reporting mechanisms underlying intervention components. Investigating the impact of protocols and role division on effective interprofessional collaboration is also crucial. Finally, this review showed varying evidence concerning different levels of PCT involvement, including being 'present', 'actively involved', or 'fully embedded'. It remains unclear what level of PCT involvement is most suitable in which context to achieve effective interprofessional collaboration contributing to improved outcomes for everyone involved.

### Implications for practice

To foster effective IPC between PCTs and ward staff, hospital decision-makers should focus on creating a supportive context that enables the necessary mechanisms identified through this realist review. Mechanisms – such as creating awareness and positive perspectives about PC among ward staff and cultivating feelings of partnership – are essential for the IPC process and eventually for delivering high-quality in-hospital PC. However, it is important to emphasize that these mechanisms are highly context-dependent, and there is no one-size-fits-all recommendation [63,64]. Each hospital operates within its own unique environment, shaped by its culture, policies, and organizational structure. Therefore, decision-makers must assess and prioritize the necessary mechanisms most relevant to their specific setting.

Drawing on insights from our third-generation CHAT analysis, which demonstrated the interconnectedness of various components within the activity systems, it is crucial to be aware of the fact no linear-causal approach exist. Instead, hospital decision-makers should adopt a holistic perspective taking into account all these different components in their organization and their reciprocal influence. This requires tailoring interventions and strategies to the dynamics within their organization, ensuring that the context supports the mechanisms necessary for driving IPC leading to integrated PC delivery within the hospital setting. Lastly, to foster a culture of learning throughout the IPC process, we recommend hospital decision-makers to implement both single-loop learning strategies, such as continuous improvements to current activities, and double-loop learning strategies, including regular reflective sessions on existing norms and values.

### Conclusion

This realist review provides valuable insights into key contextual factors triggering mechanisms related to IPC between PCTs and hospital ward staff, thereby enhancing the quality of care at both patient, relative, healthcare worker, and hospital level. Positive beliefs, attitudes, and perceptions between PC specialists and ward staff, along with clear roles and task descriptions, create beneficial conditions for effective IPC. Hospital decision-makers should prioritize raising awareness of PC across its various dimensions among ward staff and fostering a sense of partnership between PCTs and ward staff to ensure IPC and optimize high-quality PC for all hospitalized patients and their relatives with PC needs. Understanding mediating factors that influence motivation to engage in IPC is crucial for supporting organizational learning. Ultimately,

IPC between PCTs and hospital ward staff is a complex process that should be tailored to the specific needs, expectations, and norms of each and every single hospital.

## Supporting information

**S1 File. Initial CMO model.**
(PDF)

**S2 File. Search syntaxes all databases.**
(DOCX)

**S3 File. Data extraction forms.**
(DOCX)

**S4 File. General characteristics of included studies.**
(XLSX)

**S5 File. Individual CMO-configurations.**
(DOCX)

**S6 File. Recurrent mechanisms with corresponding CMO-configurations.**
(DOCX)

**S7 Fig. CHAT analysis.**
(JPG)

## Author contributions

**Conceptualization:** Louana Moons, Fouke Ombelet, Mieke Deschodt, Maaike L. De Roo, Eva Oldenburger, Inge Bossuyt, Peter Pype.

**Data curation:** Louana Moons, Fouke Ombelet.

**Formal analysis:** Louana Moons, Fouke Ombelet.

**Investigation:** Louana Moons, Fouke Ombelet, Mieke Deschodt, Peter Pype.

**Methodology:** Louana Moons, Fouke Ombelet, Mieke Deschodt, Peter Pype.

**Supervision:** Mieke Deschodt, Peter Pype.

**Validation:** Louana Moons, Fouke Ombelet, Mieke Deschodt, Maaike L. De Roo, Eva Oldenburger, Inge Bossuyt, Peter Pype.

**Visualization:** Louana Moons, Fouke Ombelet.

**Writing – original draft:** Louana Moons, Fouke Ombelet, Mieke Deschodt, Peter Pype.

**Writing – review & editing:** Louana Moons, Fouke Ombelet, Mieke Deschodt, Maaike L. De Roo, Eva Oldenburger, Inge Bossuyt, Peter Pype.

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
