## [Decision Letter · Decision Letter 0]

3 Oct 2025

Dear Dr. Deschodt,

Thank you for submitting your manuscript to PLOS ONE. After careful consideration, we feel that it has merit but does not fully meet PLOS ONE’s publication criteria as it currently stands. Therefore, we invite you to submit a revised version of the manuscript that addresses the points raised during the review process.

We look forward to receiving your revised manuscript.

Kind regards,

Suyan Tian

Academic Editor

PLOS ONE

Additional Editor Comments (if provided):

Reviewers' comments:

Reviewer's Responses to Questions

**Comments to the Author**

1. Is the manuscript technically sound, and do the data support the conclusions?

Reviewer #1: Yes

2. Has the statistical analysis been performed appropriately and rigorously?

Reviewer #1: N/A

3. Have the authors made all data underlying the findings in their manuscript fully available?

Reviewer #1: Yes

4. Is the manuscript presented in an intelligible fashion and written in standard English?

Reviewer #1: Yes

Reviewer #1: This is a well executed realist evaluation of palliative care team interaction with ward staff. Over all this was interesting to read and made some good recommendations. The CMOs are nuanced and were supported extremely well by the figures and diagrams in the Supplementary files. Interesting choice of initial program theory but it works quite well.

Introduction

More information about how the palliative care team supports the ward team should be given here. I thought initially they took over full care of the patient. A few extra sentences giving an example of a scenario where the PCT operated as intended would be helpful for readers like me who have not worked in a hospital for many years. Whose call is it to bring in the PC team? What happens? Do they keep monitoring them? How many people in the PC team?

Methods

Can you include some information about the team, elaborating on what expertise and experience was represented?

Results

I liked that you identified the prerequisites for collaboration, rather than trying to shoehorn them into the CMOs.

The Supplementary files are very useful.

Discussion

Line 501ff “To foster double-loop learning, …. By implementing such changes, interprofessional collaboration (IPC) could be improved, and ward staff would no longer feel burdened by time constraints.” Not sure this suggestion makes sense. Seems to increase the work of the ward staff rather than lessen it. Can you reconsider this paragraph? Maybe I have misunderstood and it needs clarifying.

Can you add a comment about other IPC based on this. Perhaps some elements could apply for other specialist teams? If not, can you spell out why PCT interactions are different. Justa sentence or two would do it but I was curious to see what you think.

Figure 2 is very helpful and certainly conveys the complexity of the moving parts.

A few typos or minor grammar issues

Line 89: “… as a considerate number of papers” should be “considerable”

Line 114: “…for a mechanism to get triggered” should be “for a mechanism to be triggered.”

Line 175: “(physicians, nurses, social 9 176 workers…).” Better to say (e.g., physicians, nurses, social workers)

Line 255: “non-judgement approach” should be non-judgemental

**Do you want your identity to be public for this peer review?** For information about this choice, including consent withdrawal, please see our Privacy Policy

Reviewer #1: **Yes: ** Janet C Long

---

## [Author Response · Author response to Decision Letter 1]

24 Oct 2025

Journal requirements

1. Please ensure that your manuscript meets PLOS ONE's style requirements, including those for file naming. The PLOS ONE style templates can be found at:

Thank you for bringing this to our attention. We have adjusted necessary style requirements, including those for file naming and text styles.

The reviewers’ comments did not include a recommendation to cite specific previously published

works. Hence, our reference list remained unadjusted.

Editor Comments:

None.

Reviewers’ comments:

Reviewer #1:

In general: This is a well executed realist evaluation of palliative care team interaction with ward staff. Over all this was interesting to read and made some good recommendations. The CMOs are nuanced and were supported extremely well by the figures and diagrams in the Supplementary files. Interesting choice of initial program theory but it works quite well.

Dear reviewer, thank you for taking the time to review our manuscript. We appreciate your positive feedback and are pleased you found the manuscript satisfactory. Thank you for your considerate questions and suggestions. Please find below some clarifications and adjustments we have carried out based on your feedback.

Introduction: More information about how the palliative care team supports the ward team should be given here. I thought initially they took over full care of the patient. A few extra sentences giving an example of a scenario where the PCT operated as intended would be helpful for readers like me who have not worked in a hospital for many years. Whose call is it to bring in the PC team? What happens? Do they keep monitoring them? How many people in the PC team?

Thank you very much for your thoughtful comment. We have revised the introduction to include additional detail on the composition of PCTs and added a brief example to clarify their consultative role. While PCTs typically support ward teams rather than assume full responsibility for patient care, their structure and involvement vary considerably across countries and institutions. As such, questions like “Whose call is it?” or “Do they continue monitoring?” are difficult to generalize and were not fully addressed in the introduction. Our primary aim with this paragraph was to emphasize the importance of interprofessional collaboration between PCTs and ward staff, which is central to our study.

Page 4, lines 60 – 66:

“PCTs are multidisciplinary, often including physicians, nurses and psychologist. Rather than taking over primary patient care, PCTs provide consultative support and advise to hospital ward staff regarding palliative care. For example, in a case of a patient with advanced cancer and escalating symptoms, the PCT might support hospital ward staff in advance care planning conversations, adjusting medication prescription, and coordinating family support, while hospital ward staff remains responsible for daily care delivery. This underscores the importance of effective interprofessional collaboration (IPC) to ensure optimal PC delivery [9].”

Methods: Can you include some information about the team, elaborating on what expertise and experience was represented?

We have added a description of the research team to clarify the range of expertise represented,

including clinical experience in palliative care, as well as academic expertise in implementation science, palliative care research, realist review methodology, and interprofessional collaboration.

Page 5, lines 88 – 91:

“The research team included academic experts in implementation science (MD), nursing research (LM), palliative care research (MDR), and realist review methodology and interprofessional collaboration (PP) as well as clinical experts in palliative care (EO, IB).”

Results: I liked that you identified the prerequisites for collaboration, rather than trying to shoehorn them into the CMOs. The Supplementary files are very useful.

Thank you for this positive feedback.

Discussion: Line 501ff “To foster double-loop learning, …. By implementing such changes, interprofessional collaboration (IPC) could be improved, and ward staff would no longer feel burdened by time constraints.” Not sure this suggestion makes sense. Seems to increase the work of the ward staff rather than lessen it. Can you reconsider this paragraph? Maybe I have misunderstood and it needs clarifying. Can you add a comment about other IPC based on this. Perhaps some elements could apply for other specialist teams? If not, can you spell out why PCT interactions are different. Just a sentence or two would do it but I was curious to see what you think. Figure 2 is very helpful and certainly conveys the complexity of the moving parts.

Thank you for your thoughtful comment. We agree that the original paragraph may have unintentionally implied an increased workload for ward staff. We’ve revised the section to clarify that perceived time constraints might stem from limited PC confidence rather than actual task volume. The revised paragraph now emphasizes double-loop learning as a systemic solution to reduce this sense of overwhelm. We believe that by integrating structural changes like provision of in-hospital PC training, double-loop learning could be achieved. In our opinion, this approach could empower ward staff in their generalist PC knowledge and skills, making them feel more comfortable in providing PC and feeling less overwhelmed or constrained by time, without adding to their burden.

Page 22, lines 508 – 516:

“Friction also arises when staff is confronted with a patient’s worsening clinical status but are constrained by lack of time. However, this perceived time pressure may be rooted in limited PC knowledge, leaving ward staff to feel overwhelmed, causing them to ask support from their PCT—a response characteristic of single-loop learning. To foster double-loop learning, hospitals should address these underlying causes of experiencing lack of time due to limited PC knowledge by investing in systemic solutions, for example a yearly PC training for ward staff. Such initiatives can gradually shift norms within the system and increase comfort with PC tasks among ward staff, reducing the sense of being overwhelmed even if the workload remains unchanged. While this approach may initially require additional effort, it can enhance efficiency and interprofessional collaboration (IPC) in the long term.”

A few typos or minor grammar issues:

- Line 89: “… as a considerate number of papers” should be “considerable”

- Line 114: “…for a mechanism to get triggered” should be “for a mechanism to be triggered.”

- Line 175: “(physicians, nurses, social 9 176 workers…).” Better to say (e.g., physicians, nurses, social workers)

- Line 255: “non-judgement approach” should be non-judgemental

Thank you for bringing these to our attention. We have adjusted the typos and minor grammar issues based on your feedback. In addition, I also adjusted some typos at lines 301 and 302.

---

## [Decision Letter · Decision Letter 1]

18 Nov 2025

Interprofessional collaboration between hospital-based palliative care teams and hospital ward staff: a realist review

PONE-D-25-22843R1

Dear Dr. Deschodt,

We’re pleased to inform you that your manuscript has been judged scientifically suitable for publication and will be formally accepted for publication once it meets all outstanding technical requirements.

Kind regards,

Suyan Tian

Academic Editor

PLOS ONE

Additional Editor Comments (optional):

Reviewers' comments:

Reviewer's Responses to Questions

**Comments to the Author**

Reviewer #1: All comments have been addressed

2. Is the manuscript technically sound, and do the data support the conclusions?

Reviewer #1: Yes

3. Has the statistical analysis been performed appropriately and rigorously?

Reviewer #1: N/A

4. Have the authors made all data underlying the findings in their manuscript fully available?

Reviewer #1: Yes

5. Is the manuscript presented in an intelligible fashion and written in standard English?

Reviewer #1: Yes

Reviewer #1: Thank you for your thoughtful responses. This is an excellent paper and I look forward to seeing it in print.

**Do you want your identity to be public for this peer review?** For information about this choice, including consent withdrawal, please see our Privacy Policy

Reviewer #1: **Yes: ** Janet C Long

---

## [Editor Report · Acceptance letter]

PONE-D-25-22843R1

PLOS One

Dear Dr. Deschodt,

I'm pleased to inform you that your manuscript has been deemed suitable for publication in PLOS One. Congratulations! Your manuscript is now being handed over to our production team.

Kind regards,

on behalf of

Dr. Suyan Tian

Academic Editor

PLOS One